# Leveraging dynamic stability to infer regulation in protein-protein interaction networks: A study of infectious vulnerability in COPD

Joyce Reimer[1], Jeffrey Page[2], Pranta Saha[1], Shichen Shen[3], Xiaoyu Zhu[3], Shuo Qian[3], Manoj Mammen[4], Jun Qu[3], Sanjay Sethi[5,6], Gordon Broderick[1,7,8*]

1 Vaccine and Infectious Disease Organization, University of Saskatchewan, Saskatoon, Saskatchewan, Canada, 2 Complex Exposure Threats (CETC), US Veterans Affairs, NW Washington, District of Columbia, United States of America, 3 Department of Pharmaceutical Sciences, University at Buffalo, The State University of New York, Buffalo, New York, United States of America, 4 Department of Medicine, University of Rochester Medical Center, Rochester, New York, United States of America, 5 Department of Medicine, Jacobs School of Medicine & Biomedical Sciences, University at Buffalo, The State University of New York, Buffalo, New York, United States of America, 6 VA Western New York Healthcare System, Buffalo, New York, United States of America, 7 Department of Mathematics and Statistics, College of Arts and Science, University of Saskatchewan, Saskatoon, Saskatchewan, Canada, 8 Department of Pediatrics, College of Medicine, University of Saskatchewan, Saskatoon, Saskatchewan, Canada

* gordon.broderick@usask.ca

## Abstract

The fourth leading cause of death in the US, Chronic Obstructive Pulmonary Disease (COPD) is punctuated by frequent viral and bacterial infections causing severe acute exacerbations (AECOPD) and increased mortality. In previous work we have shown that altered immune cell signaling may confer increased and persistent susceptibility to infection. Here we continue this investigation by conducting broad-spectrum proteomic profiling of circulating white blood cells to assemble an empirical protein-protein interaction network associated with frequency of infectious exacerbation. In a novel extension of conventional cross-sectional data analyses, we translate these undirected protein-protein interactions into candidate regulatory relationships with both direction and mode of action. The latter are inferred by formulating and solving a constraint satisfaction problem (SAT) whereby predicted dynamic behaviors of any valid regulatory network must support the expected persistent nature of low and high vulnerability phenotypes. Solving this SAT problem produced a set of competing candidate protein regulatory network architectures and signalling rules that unanimously highlighted several novel candidate pathway elements involved in oxidative stress response. Analysis of the overall dynamics supported by these networks, again supported the hypothesis that progression beyond an immune tipping point may confer persistent susceptibility to infection and that this may constitute a stable phenotype or regulatory trap in COPD characterized by a reactive oxygen cascade.

**Data availability statement:** All data underlying the results and source code used to conduct the analyses may be found as Supporting Information.

**Funding:** This work was supported by Rochester Regional Health in conjunction with the US Department of Defense Congressionally Directed Medical Research Programs (CDMRP) under Peer Reviewed Medical Research Program (PRMRP) award W81XWH1910804 (Broderick - PI; Sethi - Partnering PI). VIDO receives operational funding from the Canada Foundation for Innovation through the Major Science Initiatives Fund and from the Government of Saskatchewan through Innovation Saskatchewan and the Ministry of Agriculture. The funders had no role in study design, data collection and analysis, decision to publish, or preparation of the manuscript.

**Competing interests:** The authors have declared that no competing interests exist.

## Introduction

As our ability to accurately measure broad swaths of the proteome continues to evolve rapidly [1,2], often isolating numerous new proteins that have not yet been annotated. Moreover, even for those proteins that have been annotated, we frequently do not a clear understanding of their function, involvement and role in the broader highly interconnected network of intracellular pathways. To capture the context in which these proteins express, significant efforts have been directed at constructing protein-protein interactions (PPI) maps from experimental data using various measures of association and comparing their structure and connectivity patterns [3]. However, the direction in which a mediator affects a target protein and whether this action activates or inactivates the latter remains challenging to infer limiting current analyses to static comparisons of network structure and connectivity patterns [4]. This is especially true for cross-sectional observational data, which in many cases represents the majority of available data [5]. Numerical strategies have been proposed to generate artificial time course data from cross-sectional data sets that typically involve the re-ordering of pseudo- steady state observations such that they can be supported as an ordered sequence by an underlying forecasting model, e.g., Markov Chain (HM) [6,7]. Extensions of such methods have also been proposed that do not require subjects align along a single trajectory but instead that subjects with proximal profiles simply evolve in a shared direction predicted using a set of stochastic differential equations based on Langevin dynamics [8].

Here, we propose a complementary approach where instead of attempting to describe a trajectory of progression, we focus on homeostatic regulatory stability as a defining characteristic by which directed functional relationships between proteins might be inferred from standard undirected PPI maps. We apply this strategy to cross-sectional protein expression data collected as part of a larger ongoing study directed at identifying the mechanistic underpinnings that drive alternative immune responses in subjects with chronic obstructive pulmonary disease (COPD) that are especially vulnerable to infection. The fourth leading cause of death in the US, the course of COPD is punctuated by frequent acute exacerbations (AECOPD), mostly caused by tracheobronchial mucosal infection by bacteria and/or viruses [9]. Para-doxically, acute and chronic infection in COPD is prevalent despite persistent excessive inflammation and vigorous immune response to the pathogens. A 'Vicious Circle', where adaptive defects in innate immune response induced by tobacco smoke allow persistent infection, which in turn perpetuates inflammation and dysregulates innate immune response, has been proposed to explain this paradox [10]. As a potential test of this hypothesis, we investigate the role of soluble protein signaling in the peripheral circulation as an accessory to persistent inflammatory signaling in the mucosa [11] and how this might be used as a network signature of increased vulnerability to infection. In a simplifying modification of the above-mentioned methods, like subjects are expected to progress towards shared one of two stable attractors, namely the persistent phenotypes of low and high vulnerability to infection. Using a one-step discrete difference equation to describe network regulatory dynamics, our analysis

suggests that these two phenotypes can be distinguished based on the relative abundance of 15 out of 850 quantifiable protein species and that these phenotypes share a common co-regulatory network structure and state transition logic. As such it may be possible to redirect or de-escalate vulnerability to infection in highly susceptible individuals.

## Methods

### Study population and sample collection

As part of a larger ongoing study, serum samples were collected from N = 67 subjects with exacerbation history documented for up to 12 years. All subjects were males between the ages of 46–81, with a median exacerbations per year of approximately 1.8 and a median of 7 years of follow up visits (S1 File Suppl. Table S1). Most subjects presented with a GOLD symptom severity score corresponding to moderate (GOLD 2; n = 32) or severe illness (GOLD 3: n = 21). All serum samples were collected during a "well" visit at rest under conditions of stable illness. One subject corresponding to a data entry of 14 exacerbations per year was removed from analysis as a suspected outlier leaving *n* = 66 protein expression profiles (subject ID 153).

This study is a sub-study of a larger group of patients with COPD and healthy controls to understand biological determinants of exacerbation frequency and was approved by the Institutional Review Boards of the Veterans Affairs Western New York Healthcare System and University at Buffalo. The participants gave the written consent to the study via an IRB-approved consent form. The studies in this work abide by the Declaration of Helsinki principles. The biological samples and data were accessed for research purposes between 01/07/2019 and 01/05/2024. All data was de-identified and the authors did not have access to any information that could identify individual participants during or after data collection.

**Ethics statement.** This study is a sub-study of a larger group of patients with COPD and healthy controls to understand biological determinants of exacerbation frequency and was approved by the Institutional Review Boards of the Veterans Affairs Western New York Healthcare System and University at Buffalo. The participants gave the written consent to the study via an IRB-approved consent form. The participants gave the written consent to the study via an IRB-approved consent form. The studies in this work abide by the Declaration of Helsinki principles.

### Broad-spectrum LC-MS in serum

Bio-banked samples were transported to the proteomics core facility in liquid nitrogen and stored under −80°C until analysis. Prior to analysis in duplicate by the IonStar LC-MS procedure [1], samples were treated with a detergent-cocktail buffer (0.5% sodium deoxycholate, 2% SDS, 2% IGEPAL® CA-630 and protease/phosphatase inhibitor cocktail) in order to thoroughly denature the proteins and to digest efficiently the samples, followed by a surfactant-aided/precipitation-on-pellet-digestion (SOD) method to achieve quantitative and efficient recovery of peptides [2]. To achieve in-depth profiling and accurate peptide ion-current quantification, digests were separated on a 100-cm-long column with 2-μm-particles by an ultra-high-pressure chromatographic setup with optimized to deliver <15% variation of peptide signal strength [12,13]. An Orbitrap LUMOS ultra-high-field and high-resolution mass spectrometer was used to acquire quantitative ion-current signal and for protein identification. Individual data files were queried against the Swiss-Prot human protein database using the MSGF+ package. A highly stringent set of criteria (e.g., < 1% protein FDR and >2 peptides per protein) was employed to ensure detection of small (as low as 20%) changes in low abundance mediators with high confidence while also maintaining excellent depth and sensitivity (>6 orders of magnitudes in protein abundance). Using this strategy has been shown to routinely quantify 5000–6000 proteins (benchmarked in human cell lysates) in up to 150 biological samples in one batch with <1% missing data, < 10% quantitative variation (technical replicates) and <5% false-positive biomarker discovery rate benchmarked against an experimental null method described previously [1]. The original IonStar LC-MS data can be found in Supporting Information S2 File.

## Protein interaction network assembly

A protein-protein interaction network was computed and reduced to the component sub-network most relevant to the clinical phenotype according to the steps described in **Fig 1**. To eliminate bias, the abundance values were normalized to unit range for each species independently (Step 1). Interactions between range-adjusted protein species was estimated using mutual information (MI), based on equal bin width, as a more sensitive measure of association. Null model or background MI values were generated concurrently by a random shuffling of abundance values for each species repeated 50 times. At each iteration random background pairwise MI values were generated and stored as a null distribution to be used for significance testing (Step 2). Estimated MI values describing protein-protein interactions need not only be significant compared to random result but should also offer a signal of sufficient magnitude to be *meaningful* when compared to a non-random background. Here we apply Otsu's algorithm [14], typically used in image analysis, to identify a threshold MI value above which a protein-protein interaction can be considered foreground, and below which they can be considered background in the context of this data (Step 3). As a sanity check, the significance of all foreground MI values compared to a random result are verified using a Bonferroni correction for multiple comparisons enforcing a false discovery rate of less than 1% (FDR < 0.01) (Step 4). Finally all statistically significant foreground values of MI were also required to be equal or greater in magnitude than twice the average random MI (signal to noise ratio ≥2) (Step 5).

The network formed by this subset of only the most important and significant protein-protein interactions was further pruned to include only those proteins most relevant to the clinical variable of interest, namely exacerbation frequency.

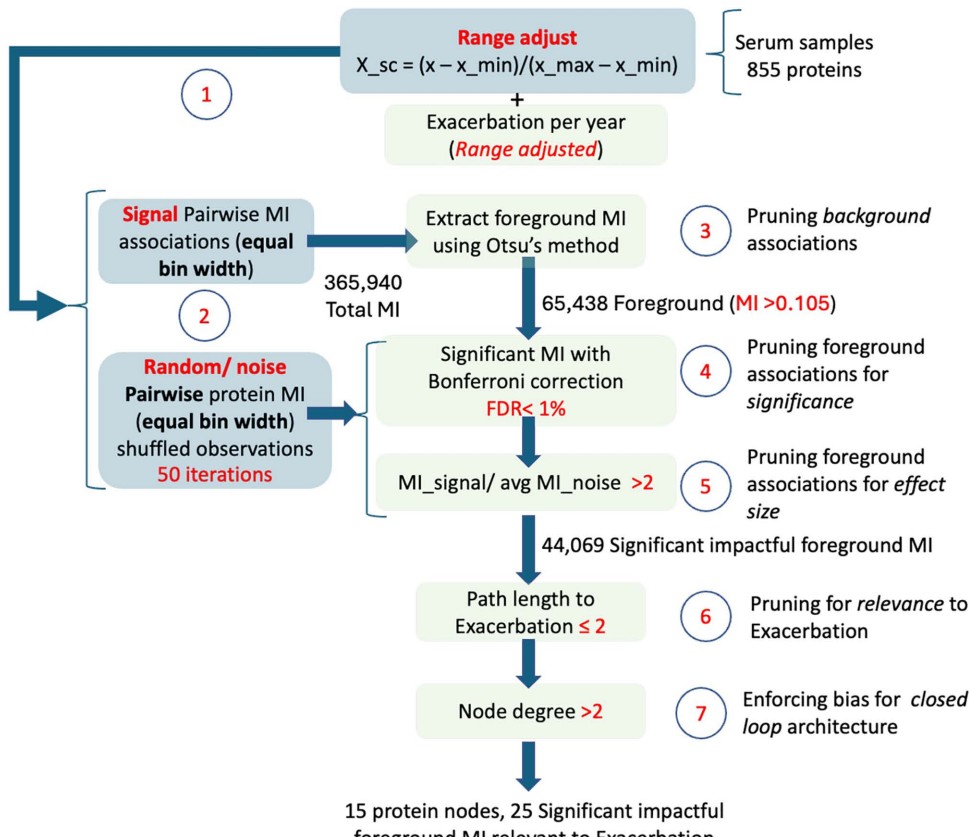

**Fig 1. Overview of data flow and analysis steps involved in the reverse engineering of a closed-loop 15-node subnetwork consisting of significant foreground protein-protein interactions most relevant to infectious exacerbation.**

Proteins indirectly related to exacerbation with a path length exceeding 2 were pruned at this stage of analysis. In other words, proteins that were first and second degree neighbors of exacerbation were retained (Step 6). Although, directionality of these relationships is unknown at this step, protein nodes in this subnetwork with a node degree of 1, cannot in principle be part of a closed loop. As we are focused here on subnetworks supporting closed loop regulatory behavior, only protein nodes with a node degree of 2 or greater were conserved (Step 7). All analyses were conducted using Python 3.12.3 libraries. Estimates of MI were computed using the *feature_selection.mutual_info_regression* routine in the scikit-learn library, with network analyses conducted using NetworkX 3.3. All gene ontology (GO) annotation and enrichment analyses were performed using the Panther database and analysis tools [15].

### Identifying molecular phenotype for infectious risk

The existence of a molecular signature in the peripheral blood that might be indicative of a predisposition to repeated infections was explored by examining similarities between subjects in terms of the joint expression of these 15 co-regulating proteins (**Fig 2**). We applied a spectral clustering technique which inherently accounts for the co-expression relationships between 15 proteins to identify 2 possible groups among the N = 66 subjects. Specifically, cluster identification was conducted using *cluster.SpectralClustering* routine from the sklearn Python library, specifying that 2 clusters be identified in a latent space of 2 eigenvectors. The Calinski-Harabasz score was used as a measure of quality of separation between groups with respect to within-group dispersion. A distribution of Calinski-Harabasz scores was also computed as a null model by assessing clusters generated by repeated random labeling of subjects. Protein co-expression profiles proposed as representative of each exacerbation phenotype were selected as those corresponding to the subject from each group with the expression profile nearest to that cluster's centroid. Once again, all computations were conducted using Python 3.12.3.

### Inferring directed protein-protein co-regulatory actions

The protein-protein interactions shown in **Fig 2** are based on a measure of association and as such are devoid of direction (source to target) and mode of action (activation or inactivation of the target node by the source node). In previous work by our group we demonstrated a method for inferring directionality and mode of action by requiring that information flow

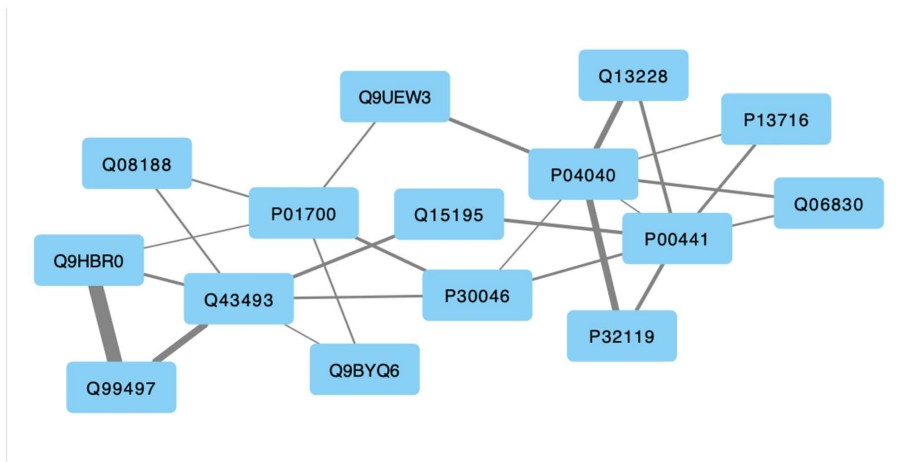

**Fig 2. Protein-protein association network (mutual information MI) consisting of 15 protein nodes linked through 25 foreground, impactful, significant MI directly or indirectly relevant to exacerbation frequency (i.e., ≤ 2 degrees of separation).** Line width is proportional to the MI value. Nodes are labelled with their UNIPROT protein identifier.

through the network enable specific observed and expected dynamic behaviors [16]. First we define at apply to each node a state transition logic whereby the next activation state of a node $s_i(t+1)$ is determined by its current state and its next logical target state or *image $s_i^*(t+1)$* as determined by the current states and actions of its upstream neighbors (Eq. 1(a)). In the current work all activation level of all node species is expressed as a ternary logic variable $s_i(t) \in [0,1,2]$ describing low, medium, or high activation levels. The distinct and sometimes competing actions of upstream neighbors, activated above their respective perception thresholds, are combined using a simple piecewise linear function. In this way, the actions of weak inactivators are weighed against those of strong activators, and vice versa, to recommend an increase or decrease in the activation of the target node be applied in the next iteration [17]. How quickly a node state actually achieves its target value is dependent on the update scheme and the maximum incremental change $\Delta s_i^{max}$ allowed.

$$s_i^*(t) = \sum_{j \in \Omega_{i=1}} w_{j,i} \; p_{j,i} \; s(t) \; // \; t_{j,i}$$

**Eq. 1(a)**

$$S_i(t+d) = S_i(t) + \min([S_i^*(t+1) - S_i(t)], \; \Delta S_i^{max})$$

**Eq. 1(b)**

The direction (source to target), mode of action $p_{j,i} \in P$ (activating or inactivating the target) and the perception threshold $t_{j,i} \in T$ for each network interaction and decisional weights $w_{j,i} \in W$ dictating each node's state transition are determined by formulating and solving a Constraint Satisfaction (SAT) problem [18,19]. As a simplification, note that the decay of a node species $s_i$ is represented implicitly by setting a basal state of zero. That is to say that should all upstream activators of a node assume a state of zero, the desired target state for that node will become zero. This parameter search problem was encoded using the open-source Constraint Programming and Modeling in Python (CPMpy) library [20] to invoke the CP-SAT solver [21] in the Google OR-Tools library [22]. CP-SAT a hybrid approach that combines finite domain propagation in Constraint Programming (CP) with the efficiency of Boolean Satisfiability (SAT) solvers. The steady state condition is articulated here as a constraint whereby the image or next target state at time $t+1$ is identical to the current state at time $t$. Additional constraints described in Table 1 include network structural considerations whereby we control for the formation of source (without upstream regulators) and sink (without any downstream target) nodes as well as nodes that are devoid of upstream activators and would invariably be downregulated to their floor state (**Table 1**). An example network structure and expression profile definition file can be found in Supporting Information S3 File. The Python source code used to parse the network definition file, state and solve the constraint satisfaction problem can be found in Supporting Information S4 File.

**Table 1. Statement of core constraints in a constraint satisfaction problem.**

| Role | Constraint |
|---|---|
| Current state $S(t)$ constrained to observed steady state $C_l$ | $s_{i,l}(t) == c_{i,l}$ for $i = 1{:}K$ state variables $l = 1{:}L$ observed steady states |
| Unnecessary edge assigned a polarity of zero implies it is also assigned a weight of zero (and vice versa) | $p_{j,i} == 0 \rightarrow w_{j,i} == 0$ $w_{j,i} == 0 \rightarrow p_{j,i} == 0$ |
| Next target state $S^*(t+1)$ is constrained to be current state $S(t)$ at each constraint $l$ | $s_{i,l}^*(t+1) == s_{i,l}(t)$ |
| Range constraints on node state as well as edge weight, polarity and threshold | $s_i \in [0,1,2]$ $w_{j,i} \in [0,1,2]$ $p_{j,i} \in [-1,0,1]$ $t_{j,i} \in [1,2]$ |
| Structural constraint avoiding formation of source (zero indegree) or sink (zero outdegree) nodes | $\sum_{j \in \Omega_{i=1}} w_{j,i} \neq 0$ |
| Controlling for only down regulation from upstream nodes | $\left| p_{j,i} \in P : p_{j,i} > 0 \right| \neq 0$ |

## Partial validation of predicted regulatory actions

In an attempt to provide a partial validation of predicted regulatory actions linking protein species in an undirected protein-protein interaction network, we conducted searches of the Elsevier Knowledge Graph database (Elsevier, Amsterdam) [23] using the EmBiology software interface and tools. This database relies on a standardized ontology where groups of synonyms describing 1.4 million biological entities are connected by 15.7 million relationships [24]. These are extracted from peer-reviewed literature using the MedScan natural language processing (NLP) engine [25,26] across multiple sources, including over 34.5 million PubMed abstracts, 430,000 clinical trials, full-text articles from 936 Elsevier journals and 939 non-Elsevier journals, as well as databases including over 200,000 entries from BioGRID, 10,000 from DrugBank and 1.3 million Reaxys drug-target relationships. The basic recognition rules and entity terminology in this ontology are updated annually with new relationships extracted from recently published PubMed. Extracted relationships are interpreted with the source exercising *Unknown, Positive, Negative, or Undefined* effects (control actions) on the target according to standardized functional processes, such as *Direct Regulation, Regulation, Expression, Protein Modification, State Change, Quantitative Change, Molecular Transport*, and others.

## Results

### A phenotypically relevant protein co-expression space

Applying the sequential selection steps described in **Fig 1** we found a candidate regulatory subnetwork consisting of 15 proteins linked by 25 significant and impactful *undirected* interactions (~12% connection density) (**Fig 2**). The strength of protein-protein associations varied between a mutual information of 0.11 and 0.54, with a median association strength of 0.16 supporting a significance of $p < 0.05$ compared to a null model of randomly sorted values (S1 File Suppl. Table S2). Of these 15 proteins, 13 were functionally annotated in Gene Ontology (GO) (Gene Ontology Consortium; release date 2024-01-17, https://doi.org/10.5281/zenodo.10536401) (S1 File **Suppl. Table S3**). Subsets of genes corresponding to these proteins were significantly over-represented in GO Biological Function classes corresponding almost unanimously to detoxification of hydrogen peroxide and cellular response stress induced by reactive oxygen species (ROS) (S1 File Suppl. Table S4(a)). Likewise, pathways curated in the Reactome database [27] that were most significantly enriched corresponded to detoxification of ROS (R-HSA-3299685) and response to chemical stress (R-HSA-9711123)(S1 File Suppl. Table S4(b)).

### Network-informed phenotypic groups

Protein abundance values for each subject were mapped onto an ordinal qualitative scale of Low = 0, Nominal = 1 and High = 2 for each species independently (S1 File Suppl. Table S5). Spectral clustering of subjects in this discrete qualitative co-expression space of 15 proteins delineated two severity groups separated by a Calinski-Harabasz score of 15.97 (p<< 0.001) (Fig 3). The first group consisted of n = 31 subjects with an average exacerbation frequency of ~2.5 episodes per year whereas the second group consisting of n = 35 subjects presented with ~1.5 exacerbation episodes per year (Fig 4). Though average exacerbations differed significantly ($p < 0.01$), the noticeable overlap in exacerbation frequencies experienced by individuals assigned to these distinct protein co-expression classes is a reminder of the important contribution of environmental factors to the clinical outcome. Subject 3 (average of 1.47 exacerbations/year) and subject 11 (average of 2.49 exacerbations/year) were identified as representative subjects for each vulnerability group by computing the centroid of each cluster and selecting the subject with the most proximal protein abundance profile. The most important differences separating the profiles for these representative subjects consist of changes in the expression of 3 proteins. Specifically, expression of catalase (EC 1.11.1.6), and delta-aminolevulinic acid dehydratase (ALADH) (EC 4.2.1.24) (Porphobilinogen synthase), were both increased in the higher risk group, while expression of D-dopachrome decarboxylase (EC 4.1.1.84) was decreased (Fig 5). In contrast, proteins involved in innate immunity were not expressed differently

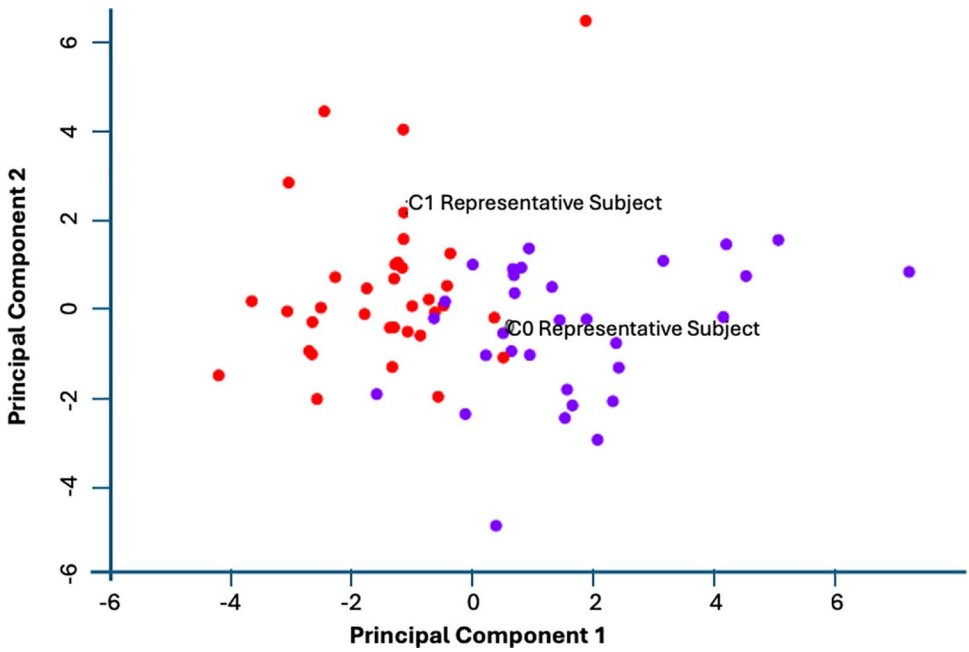

**Fig 3. Delineation of subjects into 2 clusters based on the spectral clustering of co-expression patterns in the 15 characteristic proteins shown in Fig 1.** Red dots represent n = 31 subjects with an average exacerbation frequency of ~2.5 episodes per year, whereas purple dots n = 35 subjects with ~1.5 exacerbation episodes per year. Clusters are separated by a Calinski-Harabasz score of 15.97 (p<<0.001 compared to a mean score of ~0.95 for 100 random labeling assignments).

across groups, e.g., Natural killer cell-enhancing factor A and B, Macrophage receptor with collagenous structure, etc… (S1 File Suppl. Table S3).

## Robust inference of characteristic regulatory control actions

As each of the 25 undirected protein-protein interactions was translated into a pair of opposing directed network edges, the original network architecture consisted of 50 candidate directed edges of unknown mode of action (positive or negative polarity), unknown detection threshold and unknown regulatory weight. We applied the constraints summarized in **Table 1** to a solution space spanning over ~$10^{55}$ possible combinations of parameter settings. As might be expected in such a large underdetermined parameter space, solving the corresponding SAT parameter estimation problem resulted in a large population of competing models describing low and high exacerbation profiles as persistent phenotypes equally well. However in examining a sub-sample of 100 such models we found these to be highly conserved across majority of their structure and decisional logic. Indeed, of the 150 parameters describing network structure and decisional logic, 143 are assigned the same settings in over 80 of the 100 competing models examined here, with 90 such parameter values being agreed upon unanimously (Fig 6). Details of the divergence in parameter values is presented as a phylogenetic tree diagram in supplemental figure S1 Fig. (S1 File Suppl. Table S6). A network model representative of the overall solution set could be defined as that model that is closest to the centroid of all 100 models in the parameter space. Such a representative model is presented in Fig 7, where green-colored interactions terminating in a delta arrowhead represent activation of a target by a source mediator. Likewise, red-colored interactions terminating in a T-bar indicate inactivation of a target node by an upstream mediator. An analysis of the various centrality measures for this representative model highlighted P01700 (IGLV1–47), a key factor in antibody antigen recognition, and P30046 (DDT), an inhibitor of macrophage migration, as having the highest and second highest betweenness centralities. Accordingly, their role as key information

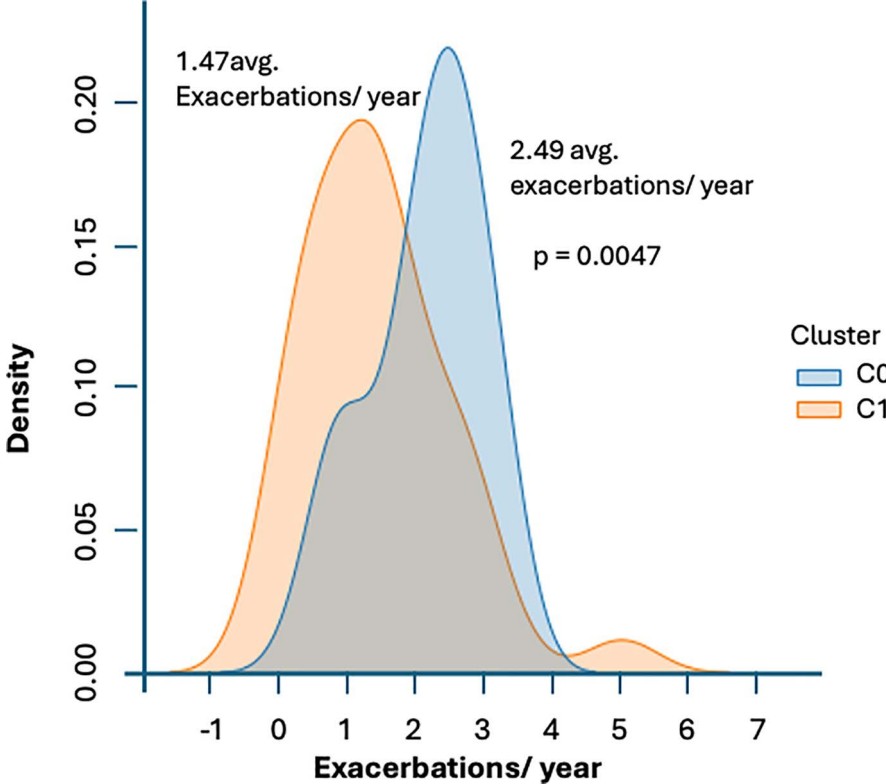

**Fig 4. Distribution of exacerbation frequencies for individuals assigned to each protein co-expression cluster from Fig 3 shows overlapping but significantly different severities on average (p<0.01).**

brokers in this network suggests dysregulation in immune recognition and innate immune response may play a role in facilitating increased vulnerability to infection in this population (S1 File Suppl. Table S7).

In a first partial verification of predicted regulatory relationships, we used the INDRA programming library [28] to interrogate the BEL Large Corpus database as well as the PathwayCommons database [29], itself an aggregate of over 23 molecular pathway databases including KEGG Pathway, Reactome, PANTHER Pathway, BioGRID, and others. Although these manually curated pathway databases did contain relationships linking some of these 15 proteins to other known regulators or targets, none contained documented relationships linking these proteins to each other. Broadening the search to also include relationships extracted from the peer-reviewed literature using automated text mining and archived in the Elsevier Knowledge Graph, we obtain 11 documented relationships linking these 15 proteins, one of which is reported in two functional categories (i.e., Regulation and Expression). The recovery of these literature informed relationships is summarized in Table 2. These 11 documented relationships are supported by a total of 66 peer-reviewed publications, 48 of these confirming a reciprocal regulatory and co-expression relationship between CAT (P04040) and SOD1 (P00441) of indeterminate action. In a majority vote across the current subset of 100 models, 8 of these 11 relationships would have been recovered correctly (~73% recall), with 10 being represented in at least a subset of models. Indeed, only in the case of documented negative co-expressive mediation of PRDX2 (P32119) by CAT (P04040) did all 100 models unanimously disagree with the 2 supporting references, predicting instead that this relationship either doesn't exist or does not contribute towards explaining our 2 representative protein abundance profiles. Conversely of the 39 potential relationships so far undocumented in the Elsevier Knowledge Graph database, 24 relationships were recruited by majority vote including 17

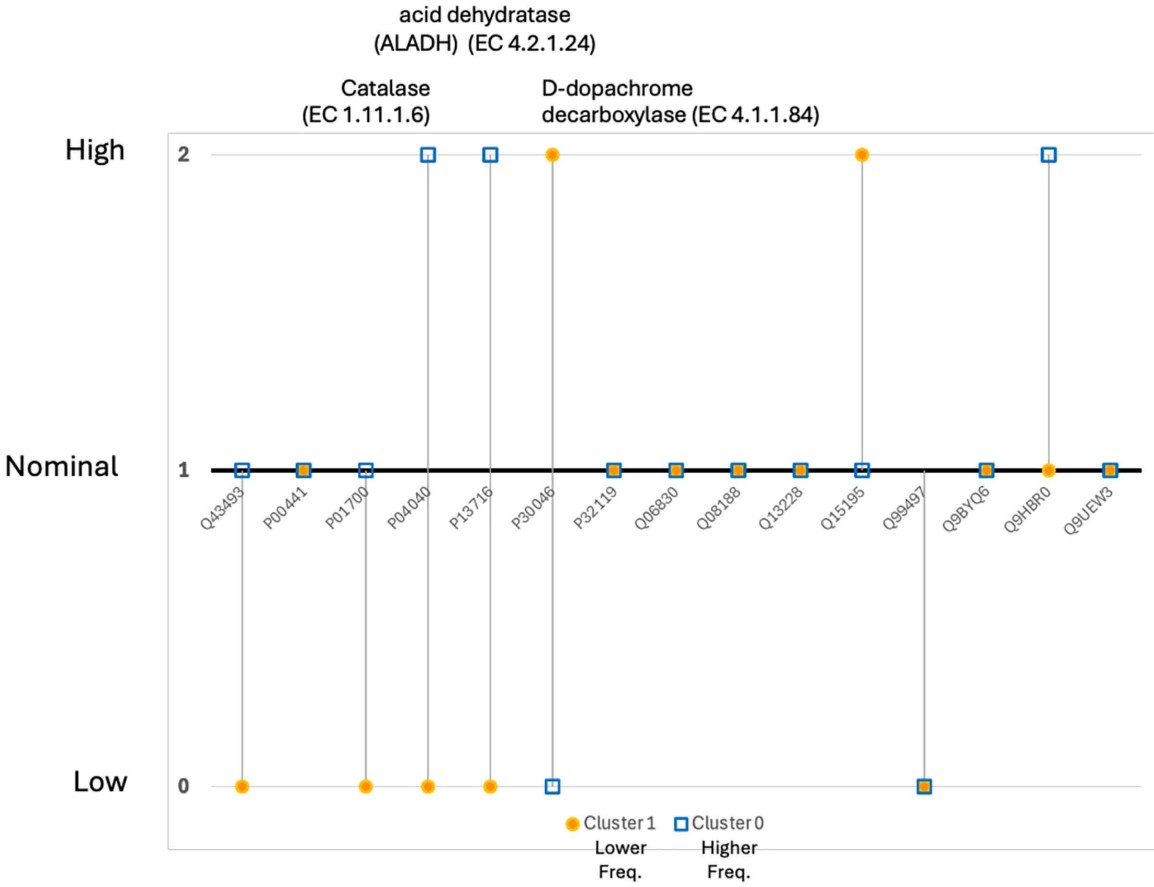

**Fig 5. Discrete ordinal scale (Low, Nominal, High) expression profiles for the subjects #3 (C1; 1.47 exacerbations/ year) and #11 (C0, 2.49 exacerbations/ year) closest to the centroid in each severity group.**

voted upon unanimously by all 100 models to explain both low and high exacerbation frequency representative profiles as persistent signatures. In a worst-case scenario where all 17 unanimously voted relationships were in fact false positives, this would equate to a positive predictive value (PPV) of roughly 32% (if we consider majority ≥ 98 models as near unanimous). Of the remaining 15 undocumented relationships, the current set of 100 models would concur in a majority vote, including 3 by unanimous vote, that these may be non-existent or at the very least not strictly required to explain the data used here as constraints. Detailed results for this model set are presented in S1 File Supplemental Table S6.

## Discussion

While network associations between markers expressed in peripheral blood have shown promise [30] as illness signatures in COPD, these have focused mainly on the identification of transcript sets. Only recently has this extended to an analysis of protein signaling in from frequent and infrequent exacerbators [31]. Though comparable in size to this study, the latter was conducted using plasma samples as opposed to sera which presents a more dilute and more subtle signature. Moreover, the analysis was again limited to the identification of tightly associated sets of differentially abundant proteins with no consideration of network structural characteristics. In this work we propose a simple numerical strategy to infer possible directed and threshold filtered control actions in an undirected protein-protein interaction network by

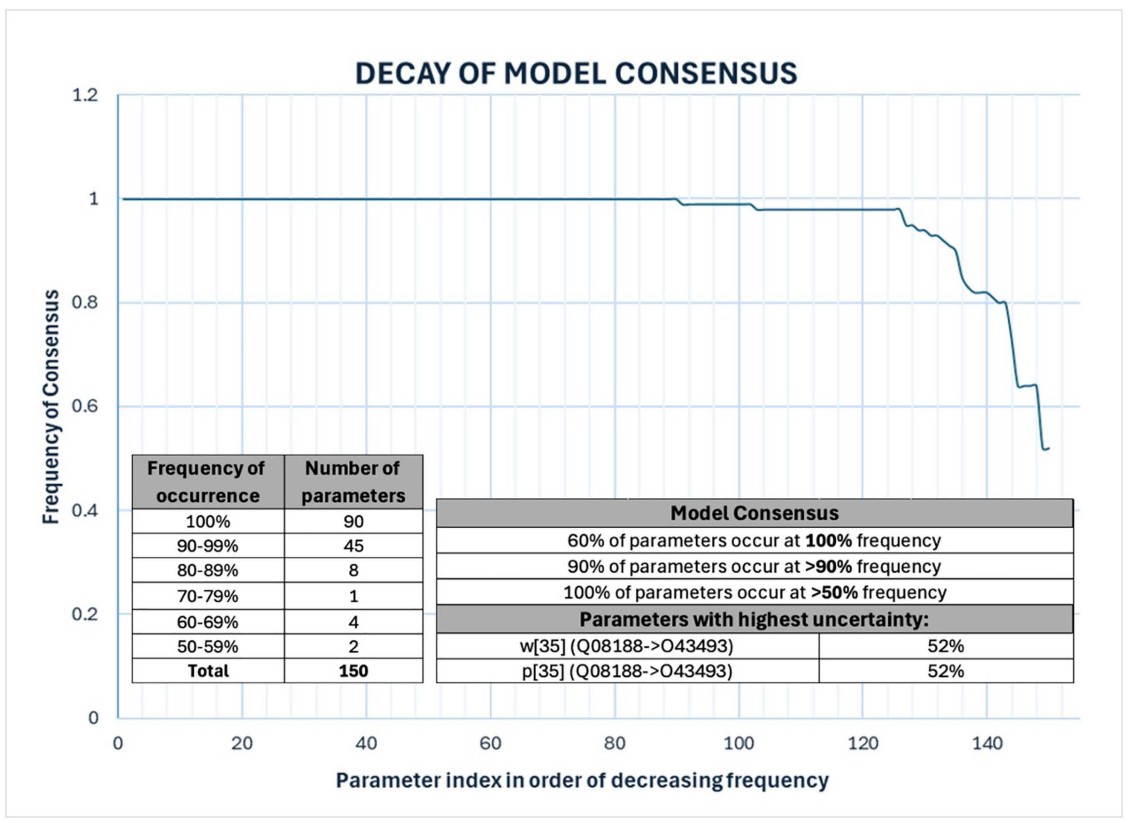

**Fig 6. Evolution in the granularity of regulatory control networks in a subset of 100 competing models describing protein abundance profiles in subjects representative of low and high exacerbation frequency as persistent phenotypes.** Parameter values describing network structure and regulatory logic are broadly conserved in over 80% of these models. The least consistent predictions involve possible reciprocal co-regulation of TGM3 (Q08188) and TGOLN2 (Q43493)(relationship 35), so far undocumented.

applying expected dynamic behaviors, in this case homeostatic stability, to cross-sectional data. We demonstrate this strategy in an analysis a high-resolution LC-MS broad-spectrum survey of serum samples collected in a larger group of n = 67 participants with stable chronic obstructive pulmonary disease (COPD) with the aim of uncovering illness-mediated changes in protein coregulation that might support increased risk of infection. In previous work by our group, an unsupervised clustering of expression profiles in the over 850 reliably identifiable protein species indicated that a subtle co-expression pattern describing as little as 2% of the variability in a subset of 160 of these proteins could delineate groups of frequent and infrequent exacerbation subjects (Calinski-Harabasz score of 160; p = 0.019). The top 15 contributors to this co-expression pattern pointed to key involvement of heme scavenging as telltale Reactome pathway element [32]. Here we revisit the selection of these candidate marker proteins from the perspective of network biology where we (i) use mutual information (MI) as a more sensitive non-linear measure of interaction, (ii) apply more stringent restrictions on the magnitude and significance of association and (iii) enforce relevance to the clinical outcome based on network proximity. Our results suggest that a small network consisting of 25 interactions linking 15 soluble protein mediators may engage in formally maintaining an altered oxidative stress response in a way that is sympathetic and perhaps mutually supportive of persistent inflammation in the mucosa described in earlier work by our group [11], with both being favorable to frequent infection. Interestingly, though there was no direct overlap in these 15 network-informed protein species with those previously identified using unsupervised clustering [32], both sets support pathway activation that predominantly aligns

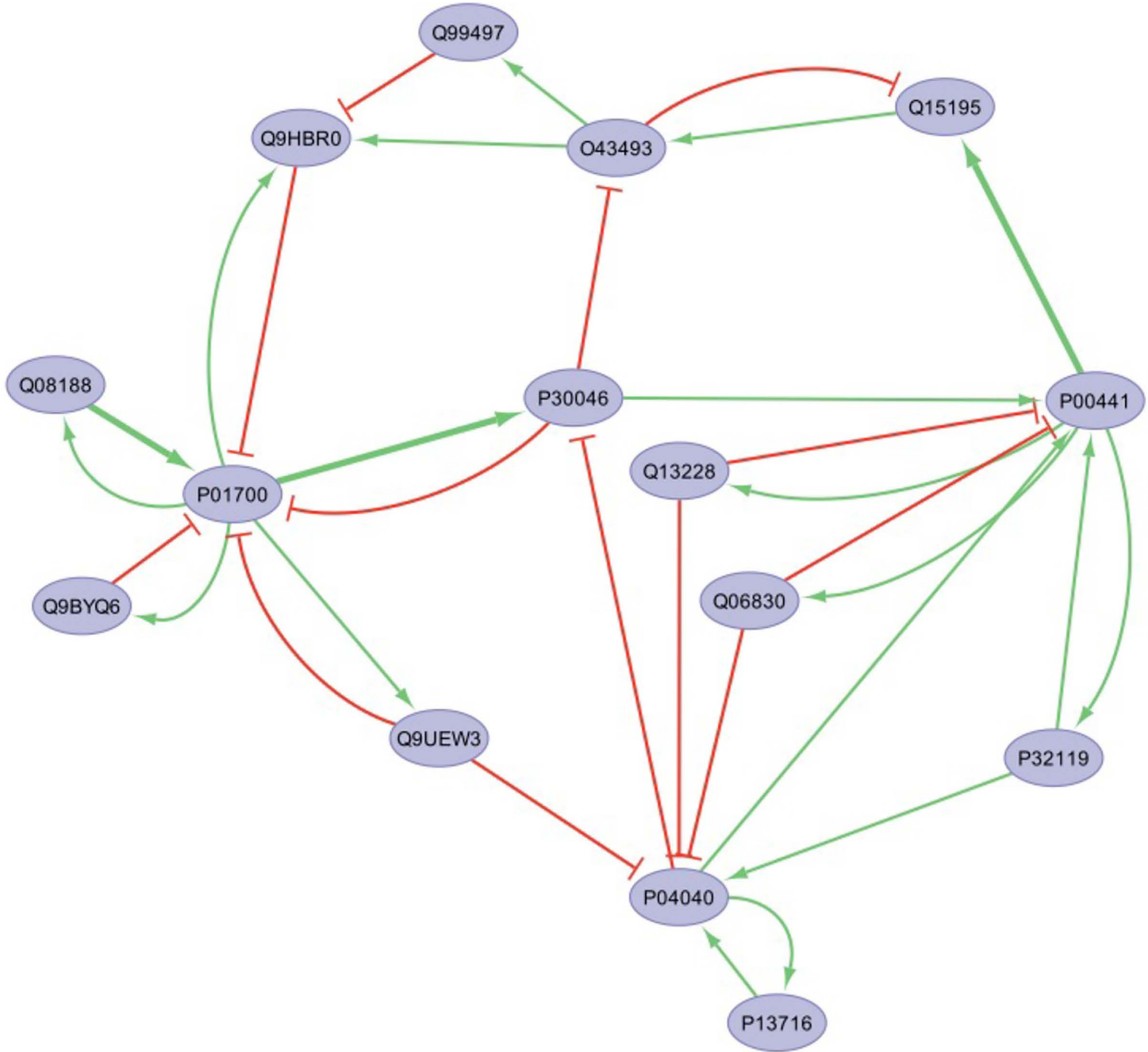

**Fig 7. A representative regulatory network, defined as that network closest to the centroid in parameter space, that explains both low and high frequency exacerbation phenotypes as persistent conditions.** Green arrows (and red T bars) represent activation (or inactivation) of a target node by an upstream mediator, with line width being proportional to the decision weight (thin w = 1, thick w = 2).

with detoxification of reactive oxygen species. Lack of agreement in individual markers as opposed to functional sets is a long-observed theme in transcriptomic analysis [33] and has been understood as a direct consequence of the interde-pendency and partial redundancy of individual molecular species in supporting a common overarching biological function. Interestingly, mediators of hemolysis and detoxification have consistently been found as markers distinguishing exposure induced COPD from autoimmune asthma [34], suggesting therapeutic avenues targeting heme scavenging [35,36]. More-over, a topological analysis of the most representative set of co-regulatory interactions highlight P01700 (IGLV1–47), a key factor in antibody antigen recognition [37], and P30046 (DDT), an inhibitor of macrophage migration [38], as being key information brokers in a potential dysregulation of immune recognition and innate immune response leading to increased vulnerability to infection in this population. It is important to note however that the important role of sex hormone mediated

**Table 2. Summary of constraint-based recovery of 11 relationships extracted from the literature using automated text mining (number of supporting references).**

| Source ID | P04040 | P00441 | P04040 | P32119 | P04040 | P30046 | Q06830 | P04040 | P13716 | Q06830 | P00441 |
|---|---|---|---|---|---|---|---|---|---|---|---|
| Target ID | P00441 | P04040 | P32119 | P04040 | P30046 | P04040 | P04040 | P13716 | P04040 | P00441 | Q13228 |
| **Model predicted** | | | | | | | | | | | |
| Polarity=−1 (Negative) | 0 | 1 | 0 | 0 | 100 | 2 | 100 | 0 | 1 | 100 | 0 |
| Polarity=0 (unsupported) | 1 | 98 | 100 | 0 | 0 | 98 | 0 | 0 | 1 | 0 | 0 |
| Polarity=+1 (Positive) | 99 | 1 | 0 | 100 | 0 | 0 | 0 | 100 | 98 | 0 | 100 |
| **Embiology** | | | | | | | | | | | |
| Source Name | CAT | SOD1 | CAT | PRDX2 | CAT | DDT | PRDX1 | CAT | ALAD | PRDX1 | SOD1 |
| Target Name | SOD1 | CAT | PRDX2 | CAT | DDT | CAT | CAT | ALAD | CAT | SOD1 | SELENBP1 |
| Direct Regulation | | | | Unknown (5) | | | | | | | |
| Regulation | Unknown (26) | Unknown (14) | | | Unknown (3) | Unknown (3) | Negative (1) | Positive (1) | Positive (1) | | |
| Expression | Unknown (8) | | Negative (2) | | | | | | | | Positive (1) |
| Modification | | | | | | | | | | Unknown (1) | |
| *Majority agreement* | Y | N | Y | Y | Y | N | YY | YY | YY | Y | YY |
| *At least one model?* | Y | Y | Y | Y | Y | Y | YY | YY | YY | Y | YY |

immune regulation was not addressed in this first study as all participants were male. This important limitation is the object of ongoing work.

Though useful in identifying functional groups [39], protein-protein interaction networks are for the most part undirected and based on associative relationships. As such they do not describe the flow of regulatory signaling. Here we identified two subjects each with a protein expression profile representative of groups enriched in subjects with a higher or lower frequency of infectious exacerbation. As these individuals were examined during stable illness, we proposed that the corresponding protein expression profile might represent a dynamically stable steady state. An earlier analysis of sputum collected in a subset of participants (n = 12) using enzyme-linked immunosorbent assays (ELISA) [11] had suggested that a common immune signaling network of documented regulatory interactions linking 10 cytokines and chemokines was capable of explaining both frequent (≤ 2 exacerbation episodes/year) and infrequent infectious phenotypes. Once again, our analysis of LC-MS protein abundance profiles representative of high and low infectious vulnerability also suggests that these could be explained by the same unifying signaling network and regulatory program. Importantly, by imposing this expected dynamic behavior on the undirected PPI network, we were able to infer not only direction but also a threshold and a mode of regulatory action for 32 of 50 potential interactions based on a majority vote across a subset of 100 competing models (~15% connection density), with 42 regulatory interactions proposed in at least a subset of models (~20% connection density). While larger simulation studies are required to more rigorously assess this approach, it was encouraging to observe that of the 11 relationships documented in the Elsevier Knowledge Graph, a broad literature and database-informed resource, 8 were recovered in a majority vote across 100 competing models with only 1 confirmed false negative. Our previous work with simulated regulatory networks indicated that low false negative rates typically aligned with high false positive rates, with a recall of ~70% typically corresponding to a positive predictive value (PPV) in the order of ~25% [40] when applying some of the more capable reverse engineering methods to perturbation time courses from the DREAM3 challenge [41] in recovering a 10-node network of 25 regulatory interactions. In this limited example, even given the worst-case scenario where all novel predicted relationships were false positives, the results presented here would still be consistent with this performance while relying on cross sectional data only. Given this important distinction, we propose that the integration of expected qualitatively described behaviors have the potential to greatly augment the usefulness of much less costly cross-sectional data sets. This framework also allows for the straightforward inclusion of general topological features as additional universal constraints [42,43], for example our requirement here that any valid network adhere to a closed loop architecture with balanced feedback (no sink nodes, no source nodes, and no strictly unipolar modulation). This has been extended in other work by our group include more complex topological features such as regulatory sub-networks or motifs [44]. Ultimately, by approaching network inference as a constraint satisfaction problem, it is possible to create hybrid initial candidate networks [24] that directly include a priori specific well-documented relationships that further constrain the choice of feasible solutions, with majority agreement in populations of models as opposed to a single unique model having the potential to deliver important improvements in both accuracy and reliability. Moreover, the identification of directed regulatory networks, each capable of supporting multiple phenotypes, is consisted with the observed multi-stability in biology underlying a variety of processes ranging from dynamic coordination in the brain [45] to cell fate decisions [46]. In a departure from the frequent focus on shifts in structure or connectivity patterns separating two separate undirected graphs, each representing a different phenotype [3,4,47], our focus here was to identify a single underlying regulatory biology capable seamlessly explaining both phenotypes as persistent or slowly progressing conditions. Such an overarching network in essence captures conventional phenotype-specific networks as simply the context-specific recruitment and dismissal of constitutive subnetworks. Importantly, the existence of a single common network and regulatory program implies that

perturbations may exist that can redirect a permissive immune homeostasis in favor of galvanizing a more robust response to infection and reduced illness severity.

## Conclusion

As our ability to broad-spectrum proteomic profiling continues to improve so does the availability of protein-protein interaction network signatures. Such networks are typically extracted from cross-sectional data and as such describe undirected associative rather than source-target co-regulatory relationships. As such, conventional analyses remain limited to a graph-analytical comparison of changes in connectivity patterns in PPI networks that might arise across phenotypic groups. Inferring the source-target direction and the regulatory action of these interactions using conventional reverse engineering methods require longitudinal time course data, often with stringent sampling requirements. Given the high cost typically involved in conducting longitudinal studies, it is useful to consider alternative approaches that might allow us to infer regulatory actions from resting state profiles, if only to provide a more informed basis for designing time course experiments. Here we propose such an alternative approach where we leverage the otherwise limiting resting state observations to ask what regulatory actions would be required of these protein-protein associations to formally satisfy this steady state condition. Though focused on a specific use case, the analysis presented in this work produced candidate regulatory networks that were broadly overlapping in structure and function. Wherever available, support in the peer-reviewed scientific literature of inferred regulatory actions was also highly favourable. Importantly, the proposed method is not intended to compete with proper time course analysis methods but instead offer an extension to current graph analytical methods when only cross-sectional data is available. In this role, we propose that additional investigation of this approach is merited.

## Supporting information

**S1 Fig. Tree diagram decomposition in the parameter space describing structural and decisional logic similarities and differences in a subset of 100 regulatory network models.**
(TIF)

**S1 File. Supplemental Tables S1 – S7** . **Table S1**. Overview of subject characteristics; **Table S2**. Protein-protein interactions based on mutual information (MI); **Table S3**. Annotation of 15 proteins of interest with Low (0), Medim (1) or High (2) expression in 2 severity clusters; **Table S4(a)**. Gene Ontology (GO) annotation in Biological Process classes – over-representation analysis of 15 proteins of interest; **Table S4(b)**. Gene Ontology (GO) annotation in Reactome Pathway classes – over-representation analysis of 15 proteins of interest; **Table S5**. Assignment of subjects into two severity clusters (high frequency = 0; low frequency = 1) based on expression of 15 proteins of interest with Low (0), Medim (1) or High (2) expression; **Table S6**. Predicted mode of action (polarity) in 100 competing models of a 15 protein co-regulatory network in serum of COPD subjects representative of two exacerbation frequency clusters (activation = +1, absent = 0, inactivation = −1); **Table S7**. Node metrics for the representative regulatory network of Fig 7.
(XLSX)

**S2 File. Original IonStar serum LC-MS proteomic profiling data.**
(XLSX)

**S3 File. Example input network structure and expression profile definition file.**
(XLSX)

**S4 File. Python source file for stating and solving the constraint satisfaction problem.**
(PY)

## Acknowledgments

The authors thank our veterans and their families for their service and their assistance with this research. Special thanks to the VA Western New York Healthcare System for their continued support of this biorepository (PI Sethi). This article is published with the permission of the Director of VIDO.

**Mandatory Disclaimer:** The opinions and assertions contained herein are the private views of the authors and are not to be construed as official or as reflecting the views of the Department of Defense.

## Author contributions

**Conceptualization:** Manoj Mammen, Jun Qu, Sanjay Sethi, Gordon Broderick.

**Data curation:** Joyce Reimer, Jeffrey Page, Shichen Shen, Xiaoyu Zhu, Shuo Qian.

**Formal analysis:** Joyce Reimer, Jeffrey Page, Gordon Broderick.

**Funding acquisition:** Manoj Mammen, Jun Qu, Sanjay Sethi, Gordon Broderick.

**Investigation:** Shichen Shen, Xiaoyu Zhu, Shuo Qian, Manoj Mammen, Jun Qu, Sanjay Sethi, Gordon Broderick.

**Methodology:** Joyce Reimer, Jeffrey Page, Manoj Mammen, Jun Qu, Sanjay Sethi, Gordon Broderick.

**Project administration:** Jun Qu, Sanjay Sethi, Gordon Broderick.

**Resources:** Pranta Saha, Shichen Shen, Xiaoyu Zhu, Shuo Qian, Manoj Mammen, Jun Qu, Sanjay Sethi, Gordon Broderick.

**Software:** Joyce Reimer, Jeffrey Page, Pranta Saha.

**Supervision:** Sanjay Sethi, Gordon Broderick.

**Validation:** Shichen Shen, Xiaoyu Zhu, Shuo Qian.

**Writing – original draft:** Joyce Reimer, Jeffrey Page, Gordon Broderick.

**Writing – review & editing:** Pranta Saha, Manoj Mammen, Jun Qu, Sanjay Sethi, Gordon Broderick.

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
