## [Decision Letter · Decision Letter 0]

15 Jul 2025

PONE-D-25-24494Leveraging Dynamic Stability to Infer Regulation in Protein-Protein Interaction Networks: A Study of Infectious Vulnerability in COPD.PLOS ONE

Dear Dr. Broderick,

Thank you for submitting your manuscript to PLOS ONE. After careful consideration, we feel that it has merit but does not fully meet PLOS ONE’s publication criteria as it currently stands. Therefore, we invite you to submit a revised version of the manuscript that addresses the points raised during the review process.

We look forward to receiving your revised manuscript.

Kind regards,

Hong Qin, PhD

Academic Editor

PLOS ONE

Journal Requirements:

“FUNDING

This work was supported by Rochester Regional Health (G. Broderick) in conjunction with the US Department of Defense Congressionally Directed Medical Research Programs (CDMRP) under Peer Reviewed Medical Research Program (PRMRP) award W81XWH1910804 (Broderick - PI; Sethi - Partnering PI). VIDO receives operational funding from the Canada Foundation for Innovation through the Major Science Initiatives Fund and from the Government of Saskatchewan through Innovation Saskatchewan and the Ministry of Agriculture (G. Broderick).”

6. Please include captions for your Supporting Information files at the end of your manuscript, and update any in-text citations to match accordingly. Please see our Supporting Information guidelines for more information: http://journals.plos.org/plosone/s/supporting-information .

Reviewers' comments:

Reviewer's Responses to Questions

**Comments to the Author**

1. Is the manuscript technically sound, and do the data support the conclusions?

Reviewer #1: Yes

Reviewer #2: Yes

2. Has the statistical analysis been performed appropriately and rigorously? 

Reviewer #1: Yes

Reviewer #2: Yes

3. Have the authors made all data underlying the findings in their manuscript fully available?

Reviewer #1: No

Reviewer #2: No

4. Is the manuscript presented in an intelligible fashion and written in standard English?

Reviewer #1: Yes

Reviewer #2: Yes

5. Review Comments to the Author

Reviewer #1: 1. This manuscript presents a novel computational approach to infer directed regulatory relationships from undirected protein-protein interaction networks using cross-sectional data from COPD patients. The work addresses an important clinical problem and introduces an innovative methodological framework. From informatics point of view, I believe this is an important work in network biology field and have shown potentially a new method of analysis this type of data.

2. However, as a reader who is interested in implementing this pipeline, I would suggest all the codes and data be made available in accordance to the open science standard. The current methods section describes the fundamental of the method but lacking the implementation or codes related to the implementation. The codes can be made available in a GitHub repository.

3. Apart from that, are the data for the LC-MS available in a database? The link to the data should be made available.

4. The availability of the entire pipeline and use case data is important in order for this analysis to be implemented on other datasets from different diseases.

Reviewer #2: Here are my comments:

1. The experiments appear to have been conducted exclusively on male subjects. This raises concerns about potential gender bias in the results. Including female subjects would have strengthened the study by improving the generalizability and robustness of the findings

2. In Figure 7, the gene network visualization displays various connections or interactions among genes. However, the meaning of the different edge colors is not explained. Additionally, are there certain genes within the network that play a more central or influential role? Clarifying these aspects would provide greater biological insight into the findings.

3. It would enhance the paper if the authors compared their proposed method against other established approaches, such as those referenced in [3, 4]. A comparative analysis would better highlight the strengths and potential limitations of the proposed method.

4. All figures should be positioned close to their corresponding references in the main text. As currently presented, it is difficult to follow the narrative without frequently scrolling between the text and the figures. This disrupts the flow of reading and makes it harder to interpret the results effectively.

5. Several figures, particularly Figure 3, lack a proper legend. Including clear legends is essential for readers to understand the plotted information without ambiguity.

6. PLOS authors have the option to publish the peer review history of their article (what does this mean? ). If published, this will include your full peer review and any attached files.

**Do you want your identity to be public for this peer review?** For information about this choice, including consent withdrawal, please see our Privacy Policy .

Reviewer #1: No

Reviewer #2: No

---

## [Author Response · Author response to Decision Letter 1]

16 Jul 2025

Reviewer #1:

1. This manuscript presents a novel computational approach to infer directed regulatory relationships from undirected protein-protein interaction networks using cross-sectional data from COPD patients. The work addresses an important clinical problem and introduces an innovative methodological framework. From informatics point of view, I believe this is an important work in network biology field and have shown potentially a new method of analysis this type of data.

Ans.

Sincere thanks for the kind words.

2. However, as a reader who is interested in implementing this pipeline, I would suggest all the codes and data be made available in accordance to the open science standard. The current methods section describes the fundamental of the method but lacking the implementation or codes related to the implementation. The codes can be made available in a GitHub repository.

Ans.

We very much agree and apologize for this omission. We have now added as new supplementary material the Python source code ( COPD_cpmpy_param_est_from_list.py) that will read a sample input file (COPD_Serum_15_PPI_input.xlsx) and produce a set of competing solutions consisting of directed networks with corresponding logic parameter values. We now point to these files in the Methods section. All such changes to the main text have been highlighted in blue font for the convenience of the reviewer.

3. Apart from that, are the data for the LC-MS available in a database? The link to the data should be made available.

Ans.

Again, sincere apologies for this oversight. We have now also provided as supplementary material the original LC-MS data (COPD-proteomics_serum_original_data.xlsx).

4. The availability of the entire pipeline and use case data is important in order for this analysis to be implemented on other datasets from different diseases.

Ans.

We agree with the reviewer and have amended the main text pointing to these new supplementary files. All such changes to the text have been highlighted in blue font for the convenience of the reviewer.

Reviewer #2: Here are my comments:

1. The experiments appear to have been conducted exclusively on male subjects. This raises concerns about potential gender bias in the results. Including female subjects would have strengthened the study by improving the generalizability and robustness of the findings.

Ans.

We agree with the reviewer. Unfortunately, the scope of the current project as funded by the US DoD directed sex comparisons as part of a future phase of work. We have emphasized this limitation in the Discussion and highlighted changes to the text in blue font for the convenience of the reviewer.

2. In Figure 7, the gene network visualization displays various connections or interactions among genes. However, the meaning of the different edge colors is not explained. Additionally, are there certain genes within the network that play a more central or influential role? Clarifying these aspects would provide greater biological insight into the findings.

Ans.

While the color and type of interaction were described in the caption for Figure 7, we agree with the reviewer and have modified the main text to clarify this further. Regarding the relative importance of the genes in the representative model of Figure 7, we very much agree with the reviewer and appreciate the reviewer raising this point. We have now conducted a topological analysis of the network on Figure 7 and have include these results in the new Supplemental Table S7. It is interesting to note P01700 (IGLV1-47), a key factor in antibody antigen recognition [Huang et al., 2024], and P30046 (DDT), an inhibitor of macrophage migration [Merk et al., 2011], as having the highest and second highest betweenness centralities highlighting potential dysregulation in immune recognition and innate immune response as playing a key role in increased vulnerability to infection. We have inserted a new passage to this effect in Section 3.3 and to the Discussion.

Merk M, Zierow S, Leng L, Das R, Du X, Schulte W, Fan J, Lue H, Chen Y, Xiong H, Chagnon F. The D-dopachrome tautomerase (DDT) gene product is a cytokine and functional homolog of macrophage migration inhibitory factor (MIF). Proceedings of the National Academy of Sciences. 2011 Aug 23;108(34):E577-85.

Huang X, Xiong L, Zhang Y, Peng X, Ba H, Yang P. Proteomic profile of the antibody diversity in circulating extracellular vesicles of lung adenocarcinoma. Scientific Reports. 2024 Nov 14;14(1):27953.

Again, all changes to the main text have been highlighted in blue font for the convenience of the reviewer.

3. It would enhance the paper if the authors compared their proposed method against other established approaches, such as those referenced in [3, 4]. A comparative analysis would better highlight the strengths and potential limitations of the proposed method.

Ans.

We very much appreciate this suggestion. The methods presented in references [3] and [4] consist in comparing two separate undirected graphs, each representing a different phenotype, and describing shifts in their structure or connectivity patterns. Our group has performed such analyses in the past (e.g. Broderick et al., 2010) and we could have identified separate graphs for the low and high exacerbation frequency phenotypes in this instance as well. However, the objective was to explore the feasibility of identifying a single regulatory network that would support both phenotypes as persistent or slowly progressing conditions. The existence of a single common network and regulatory program supporting both phenotypes implies that perturbation to this network exist that can redirect a network behavior from a high frequency response back to response patterns characteristic of a lower illness severity. We propose to the reviewer that this is significantly more challenging to articulate in terms of shifts in active interactions when working with separate often undirected interaction graphs. We have added a statement to this effect in the Discussion (please see insertions in blue font).

Broderick G, Fuite J, Kreitz A, Vernon SD, Klimas N, Fletcher MA. A formal analysis of cytokine networks in chronic fatigue syndrome. Brain, behavior, and immunity. 2010 Oct 1;24(7):1209-17.

4. All figures should be positioned close to their corresponding references in the main text. As currently presented, it is difficult to follow the narrative without frequently scrolling between the text and the figures. This disrupts the flow of reading and makes it harder to interpret the results effectively.

Ans.

We agree with the reviewer and apologize for the awkward generic format of the draft manuscript submission. It is our understanding that this will be addressed during typesetting should the manuscript be accepted for publication.

5. Several figures, particularly Figure 3, lack a proper legend. Including clear legends is essential for readers to understand the plotted information without ambiguity.

Ans.

We agree and have added additional explanations to the caption of Figure 3 as well as annotated the figure itself, now uploaded as Figure_3_r1. We apologize for this oversight and appreciate the reviewer’s patience.

---

## [Decision Letter · Decision Letter 1]

11 Aug 2025

Leveraging Dynamic Stability to Infer Regulation in Protein-Protein Interaction Networks: A Study of Infectious Vulnerability in COPD.

PONE-D-25-24494R1

Dear Dr. Broderick,

We’re pleased to inform you that your manuscript has been judged scientifically suitable for publication and will be formally accepted for publication once it meets all outstanding technical requirements.

Kind regards,

Hong Qin, PhD

Academic Editor

PLOS ONE

Additional Editor Comments (optional):

Reviewers' comments:

Reviewer's Responses to Questions

**Comments to the Author**

1. If the authors have adequately addressed your comments raised in a previous round of review and you feel that this manuscript is now acceptable for publication, you may indicate that here to bypass the “Comments to the Author” section, enter your conflict of interest statement in the “Confidential to Editor” section, and submit your "Accept" recommendation.

Reviewer #1: All comments have been addressed

Reviewer #2: All comments have been addressed

2. Is the manuscript technically sound, and do the data support the conclusions?

Reviewer #1: Yes

Reviewer #2: Yes

3. Has the statistical analysis been performed appropriately and rigorously? 

Reviewer #1: Yes

Reviewer #2: Yes

4. Have the authors made all data underlying the findings in their manuscript fully available?

Reviewer #1: Yes

Reviewer #2: Yes

5. Is the manuscript presented in an intelligible fashion and written in standard English?

Reviewer #1: Yes

Reviewer #2: Yes

6. Review Comments to the Author

Reviewer #1: My concerns have been addressed. The access to data and codes will help the field and allow other groups to replicate the work onto other datasets. Thank you.

Reviewer #2: The authors addressed all of my concerns I raised before. They did additional experiments and added new plots to answer my questions. They have used topological data analysis techniques to find out the important genes from a network. I want them to add the full description of the TDA technique they used in this version.

7. PLOS authors have the option to publish the peer review history of their article (what does this mean? ). If published, this will include your full peer review and any attached files.

**Do you want your identity to be public for this peer review?** For information about this choice, including consent withdrawal, please see our Privacy Policy .

Reviewer #1: No

Reviewer #2: No

---

## [Editor Report · Acceptance letter]

PONE-D-25-24494R1

PLOS ONE

Dear Dr. Broderick,

I'm pleased to inform you that your manuscript has been deemed suitable for publication in PLOS ONE. Congratulations! Your manuscript is now being handed over to our production team.

Kind regards,

on behalf of

Dr. Hong Qin

Academic Editor

PLOS ONE